# Mendelian Randomization Reveals Potential Causal Relationships Between DNA Damage Repair-Related Genes and Inflammatory Bowel Disease

**DOI:** 10.3390/biomedicines13010231

**Published:** 2025-01-19

**Authors:** Zhihao Qi, Quan Li, Shuhua Yang, Chun Fu, Burong Hu

**Affiliations:** 1School of Public Health, Wenzhou Medical University, Wenzhou 325035, China; qzh961204@gmail.com (Z.Q.); liquan808495@gmail.com (Q.L.);; 2Zhejiang Engineering Research Center for Innovation and Application of Intelligent Radiotherapy Technology, The Second Affiliated Hospital of Wenzhou Medical University, Wenzhou 325000, China

**Keywords:** DNA damage repair, inflammatory bowel disease, integrative omics, Mendelian randomization

## Abstract

DNA damage repair (DDR) plays a key role in maintaining genomic stability and developing inflammatory bowel disease (IBD). However, no report about the causal association between DDR and IBD exists. Whether DDR-related genes are the precise causal association to IBD in etiology remains unclear. Herein, we employed a multi-omics summary data-based Mendelian randomization (SMR) approach to ascertain the potential causal effects of DDR-related genes in IBD. **Methods:** Summary statistics from expression quantitative trait loci (eQTL), DNA methylation QTL (mQTL), and protein QTL (pQTL) on European descent were included. The GWAS summarized data for IBD and its two subtypes, Crohn’s disease (CD) and ulcerative colitis (UC), were acquired from the FinnGen study. We elected from genetic variants located within or near 2000 DDR-related genes in cis, which are closely associated with DDR-related gene changes. Variants were selected as instrumental variables (IVs) and assessed for causality with IBD and its subtypes using both SMR and two-sample MR (TSMR) approaches. Colocalization analysis was employed to evaluate whether a single genetic variant simultaneously influences two traits, thereby validating the pleiotropy hypothesis. **Results:** We identified seven DDR-related genes (*Arid5b*, *Cox5a*, *Erbb2*, *Ube2l3*, *Gpx1*, *H2bcl2*, and *Mapk3*), 33 DNA methylation genes, and two DDR-related proteins (CD274 and FCGR2A) which were all causally associated with IBD and its subtypes. Beyond causality, we integrated the multi-omics data between mQTL-eQTL and conducted druggability values. We found that DNA methylation of *Erbb2* and *Gpx1* significantly impacted their gene expression levels offering insights into the potential regulatory mechanisms of risk variants on IBD. Meanwhile, CD247 and FCGR2A could serve as targets for potential pharmacological interventions in IBD. **Conclusions:** Our study demonstrates the causal role of DDR in IBD based on the data-driven MR. Moreover, we found potential regulatory mechanisms of risk variants on IBD and potential pharmacological targets.

## 1. Background

Inflammatory bowel disease (IBD), encompassing two main subtypes, Crohn’s disease (CD) and ulcerative colitis (UC), represents a global health challenge due to its escalating incidence [1], and they are also associated with an elevated risk of colorectal cancer (CRC) [2]. It is estimated that over 3 million individuals in the USA and Europe are affected by IBD, with its prevalence in North America, Oceania, and several European countries projected to surpass 0.3% [3]. Studies have elucidated the multifaceted pathogenesis of IBD, implicating genetic susceptibility, environmental factors, intestinal microbiota, and immune dysregulation [4]. However, the results generated from most studies do not exhibit consistency in the precise etiology of IBD, and the methodologies utilized often overlook the influence of confounders, making it challenging to distinguish cause from consequence [5]. Therefore, the etiology of IBD remains largely elusive.

DNA damage repair (DDR), pivotal for recognizing and rectifying DNA lesions induced by endogenous and exogenous agents, plays a fundamental role in maintaining DNA integrity. Accumulating unrepaired DNA damages poses a risk for carcinogenesis and other disorders [6,7]. Recent evidence suggests a close association between DDR and IBD. Activation of the DDR pathway triggers sustained inflammatory responses, including cytokine release and cellular cytotoxicity, constituting the pathological basis of IBD [8,9]. Therefore, the DDR pathway’s dysfunction exacerbates inflammation and tissue damage in the gastrointestinal tract. Additionally, DDR-related molecules such as p53, CHK1, BRCA, and H2AX are implicated in inflammation-associated senescence, exacerbating the progression of IBD [10]. While the findings from these studies underscore the significant role of DDR in IBD, the etiologically specific DDR-related molecules remain unclear.

Genome-wide association studies (GWASs), leveraging single nucleotide polymorphisms (SNPs) to probe genetic association, have proven instrumental in identifying diseases-related loci [11]. Mendelian randomization (MR), a statistical approach employing genetic variants as instrumental variables, can be applied to facilitate causal inference from summary-level GWAS data [12]. The MR method has also been used in IBD research. A summary-data-based MR analysis indicates causal associations between mitochondrial-related genes and IBD [5]. Moreover, a proteome-wide MR investigation identifies circulating proteins MST1, CXCL5, and STAT3 as potential therapeutic targets for IBD [13]. With the expanding application of large-scale GWAS and QTL data, further exploring the causal relationship between DDR-related genes and IBD is viable and warranted.

Here, we first investigated potential causal associations between DDR gene expression, DNA methylation, and protein expression levels in IBD utilizing SMR or TSMR. Subsequently, we integrated multi-omic data between mQTL and eQTL to reveal potential regulatory mechanisms of risk variants on IBD. Finally, we evaluated the potential druggability of the identified DDR genes.

## 2. Methods

This study followed essential principles from the Strengthening the Reporting of Observational Studies in Epidemiology (STROBE) guidelines, particularly focusing on the guidelines related to Mendelian randomization (MR) (Appendix A) [14].

### 2.1. Study Design

A comprehensive overview of the study design is shown in Figure 1, including the detailed workflow for selecting genetic variants and the analytical methodologies employed. The genes related to DDR were sourced from the GeneCards database (https://www.genecards.org/, (accessed on 6 May 2024)) by searching for the term “DNA damage repair” and filtering with a relevance score of 8 or higher. In this study, we acquired instrumental variables representing DDR-related genes by measuring methylation, gene expression, and protein abundance. Subsequently, Mendelian randomization (MR) analysis was conducted to investigate the causal relationship between DDR-related genes and IBD as well as its subtypes at each biological level. Colocalization and sensitivity analyses were then performed to enhance the strength of causal inference. We then conducted a phenome-wide scan to detect the pleiotropy and specificity of our instrumental variables and enhance the feasibility of causal inference. Finally, we evaluated druggability of DDR genes. The exposure and outcome populations are derived from distinct study cohorts.

### 2.2. Data Sources of mQTL, eQTL and pQTL

Quantitative Trait Loci (QTL) analysis plays a crucial role in genetics and molecular biology by identifying associations between SNPs and variations in DNA methylation, gene expression, and protein abundance (Appendix A). This approach is pivotal in understanding the genetic basis of intricate traits and diseases. The eQTL data were acquired via the eQTL Gene Consortium, which meticulously compiled information on 10,317 SNPs linked to attributes in a cohort of 31,684 individuals. In our study, we selected genetic variants demonstrating strong associations with gene expression within the cis-region spanning 1 Mb as instrumental variables for genes implicated in DDR [15]. The mQTL data were obtained from an analysis conducted by McRae et al., encompassing a cohort of 1980 individuals with European ancestry. These findings elucidate the impact of SNPs at specific genomic loci on methylation levels at proximal CpG sites [16]. The data of circulating cis-pQTL were available from a pQTL research conducted by Ferkingstad et al., including 35,559 individuals of Icelandic descent [17]. DDR-related genes in the mQTL, eQTL, and pQTL datasets were separately determined. After conducting a comprehensive screening for DDR-related genes, we identified 5217 CpG-SNP pairs, 1625 eGen-SNP pairs, and 712 protein-SNP pairs (*p* < 5 × 10^−8^), respectively.

### 2.3. IBD Outcome Datasets

Our study utilized aggregated data from the FinnGen study [18]. GWAS data on IBD and its subtypes were obtained from public R10 data released from the FinnGen study, including 9083 patients with IBD and 403,098 controls, 2205 patients with CD and 392,974 controls, and 5931 patients with UC and 405,386 controls (the period of this study was 2017–2023 ) (Appendix A).

### 2.4. SMR Analysis

The SMR method, an augmentation of the MR framework, relies on three key assumptions: the relevance assumption, which requires that selected genetic variants (such as single nucleotide polymorphisms, or SNPs) are significantly associated with the exposure of interest; the independence assumption, which necessitates that these genetic variants are independent of all potential confounders, ensuring their effect on the outcome is solely mediated through the exposure; and the exclusion restriction assumption, which states that genetic variations affect the outcome exclusively through the exposure, without direct influence through other pathways. These assumptions together ensure the reliability and validity of causal inferences in Mendelian randomization studies. The SMR method was developed to quantify associations between genetically determined traits and complex traits (such as disease phenotypes) [19]. In this study, SMR was performed to identify the association between DDR-related genes and the risk of IBD. The SMR software (SMR v1.3.1) for Linux was utilized to perform SMR analysis through the command line, employing default options. The statistical power of SMR was significantly enhanced compared to traditional MR analyses, particularly when data were drawn from two separate large samples. The window (±100 kb) centered on each gene was selected to capture genetic changes closely associated with gene activity. Subsequently, the top relevant cis-QTL were chosen based on a significance threshold of 5 × 10^−8^, and their impact on gene expression and disease risk was investigated. To ensure consistency and reliability of the study findings, SNPs with allele frequency differences exceeding 0.2 among different populations were excluded from the analysis. To differentiate pleiotropy from linkage effects, we employed the Heterogeneity in Dependent Instruments (HEIDI) test. SNPs with *P*-HEIDI values less than 0.05 were considered indicative of potential pleiotropy and thus excluded from further analysis. The Benjamini–Hochberg (BH) approach was employed to adjust the *p*-value, ensuring control over the false discovery rate (FDR). The significance threshold was established at a value of α = 0.1. When the FDR-adjusted *p*-value < 0.1 and *P*-HEIDI > 0.05, colocalization analysis was performed.

### 2.5. TSMR Analysis

pQTL data were analyzed in conjunction with GWAS data using five MR methods: MR Egger, weighted median, IVW, simple mode, and weighted mode. Pleiotropy and heterogeneity were assessed with MR Egger intercept tests and leave-one-out analysis to ensure the robustness of the results. All analyses were performed in R (version 4.3.2).

### 2.6. Colocalization Analysis

Colocalization analysis plays a crucial role in strengthening associations observed in MR analysis by employing a series of arithmetic operations and statistical tests to determine the likelihood of observed overlap or spatial proximity being due to chance. This analytical approach aids in identifying MR associations that may be influenced by linkage disequilibrium (LD), where another genetic variant in high LD with the instrumental variable could also impact outcomes. In our study, we utilized a Bayesian framework for colocalization analysis of two traits, considering posterior probabilities (PP) greater than 0.7 as substantial evidence of colocalization [20]. Through the utilization of the color R package, statistical analyses were carried out.

### 2.7. Phenome-Wide Scan Analysis

We performed a phenome-wide scan with LDtrait to examine the relationship between identified QTL and other traits, accounting for confounding factors. SNPs are deemed pleiotropic under the following criteria: (1) the association has genome-wide significance (*p* < 5 × 10^−8^); (2) the SNPs shared the same effect allele with our results; (3) the absolute value of size effect (β) > 0.01.

### 2.8. Identification of Druggable Protein Targets

Evaluating protein–drug interactions is essential for assessing the viability of a target protein as a potential therapeutic target. We conducted an extensive search on DrugBank, systematically gathering information on diverse proteins and performing a comprehensive evaluation to validate their viability as prospective drug targets [21]. During this process, we categorized the identified proteins into two groups: (1) approved (indicating that one or more drugs have received approval for targeting a specific protein); (2) proteins not currently listed as therapeutic agents (i.e., proteins that have not been utilized in therapy). This classification aids in comprehending the relationship between each protein and existing drugs, as well as their potential role in future drug development.

## 3. Results

### 3.1. SMR and Colocalization Analysis Identifies DDR-Related Genes Associated with IBD

After conducting SMR analysis, we identified a significant association between the expression of 1625 DDR-related genes and IBD (Figure 1). An uncorrected *p*-value can result in a substantial number of false positives, erroneously identifying associations. By employing FDR correction, we adjusted the *p*-value to effectively control the false positive rate, ensuring enhanced reliability of the identified significant associations. An FDR-adjusted *p*-value < 0.1 signifies compelling evidence of an association. At the same time, when PHEIDI > 0.05, it indicates that these associations are not caused by pleiotropy. To further mitigate confounding arising from linkage disequilibrium (LD), we conducted a colocalization analysis. A PPH4 value exceeding 0.70 provides robust support for a shared causal variant between eQTL and IBD GWAS. After multiple testing corrections, a HEIDI test, and colocalization analyses, the genetic predictions of *Cox5a* expression were observed to be significantly associated with an increased risk of IBD (OR 1.773, 95% CI 1.284–2.263; PPH4 = 0.867). Similarly, a strong association was detected between *Ube2l3* expression and IBD susceptibility (OR 1.103, 95% CI 1.056–1.151; PPH4 = 0.908). Meanwhile, *Arid5b* (OR 0.835, 95% CI 0.756–0.922; PPH4 = 0.845) and *Erbb2* (OR 0.343, 95% CI 0.202–0.582; PPH4 = 0.921) expression could decrease the IBD risk. For UC, a genetically predicted increase in one SD in *Arid5b* (OR 0.742, 95% CI 0.656–0.84; PPH4 = 0.795), *Gpx1* (OR 0.19, 95% CI 0.127–0.294; PPH4 = 0.868), *H2bc12* (OR 0.852, 95% CI 0.781–0.93; PPH4 = 0.774), and *Mapk3* (OR 0.884, 95% CI 0.822–0.95; PPH4 = 0.898) expression could decrease the risk (Figure 2 and Appendix A).

### 3.2. SMR Analysis Identifies DDR-Related Gene Methylation Associated with IBD

Appendix A demonstrate the causal impact of DNA methylation in DDR-related genes on IBD and its subtypes. After FDR correction and HEIDI test, we detected 200 CpG sites close to 117 distinct genes. Specifically, we found 97 CpG loci within 61 unique genes associated with IBD, 23 CpG loci within 12 unique genes associated with UC, and 80 CpG loci within 44 unique genes associated with CD. Through colocalization analysis (PPH4 > 0.70), a total of 33 unique genes and 65 CpG loci were strongly supported by colocalization evidence (Appendix A).

### 3.3. SMR Analysis Identifies DDR-Related Proteins Associated with IBD

Based on the results of SMR analysis, and after FDR correction and HEIDI test, we further confirmed that the causal relationship between DDR-associated proteins and IBD was caused by shared causal variables rather than pleiotropy. There were 11 DDR-related proteins associated with IBD risk. Additionally, 9 DDR-related proteins were associated with UC risk (Appendix A). To further investigate causal relationships between DDR-related proteins and IBD, we used five MR analysis methods (MR Egger, weighted median, IVW, simple mode, and weighted mode approaches). Additionally, we used the intercept of MR Egger to test for horizontal pleiotropy, and the results indicated no horizontal pleiotropy (*p*-value > 0.05) (Appendix A). Through colocalization analysis, we found that increased expression of CD274 (OR 0.77, 95% CI 0.67–0.89; PPH4 = 0.875) and FCGR2A (OR 0.90, 95% CI 0.88–0.92; PPH4 = 0.982) reduced the risk of IBD; meanwhile, FCGR2A (OR 0.88, 95% CI 0.85–0.90) also had a protective effect against UC (Figure 3).

### 3.4. Exploring the Causal Relationship Between DDR-mQTL and DDR-eQTL

To investigate the impact of DDR-related gene methylation on expression, we conducted SMR analysis. This process involved identifying shared genetic variants influencing both gene methylation and expression, followed by rigorous corrections for multiple testing and HEIDI tests. As a result, we compiled a gene list where DNA methylation CpG sites regulate DDR gene expression. Based on identified DDR-related genes associated with IBD, we confirmed associations: rs2517953 regulates *Erbb2* gene methylation linked to *Erbb2* expression, and rs11903592 and rs4241406 regulate *Gpx1* gene methylation associated with *Gpx1* expression (Table 1 and Appendix A).

### 3.5. Phenome-Wide Scan Analysis of Genetic Variants to Detect Reverse Causality

To exclude potential pleiotropic effects of SNPs, we identified and ensured the accuracy and reliability of causal relationships with IBD. We conducted a phenome-wide scan using LDtrait on genetic variants [22]. Phenome-wide scan results are shown in Appendix A. Certain genes were linked to established secondary traits: rs2070512 (*Ube2l3*-expression associated) and rs3807307 (*Irf5*-methylation associated) were associated with secondary traits such as total fatty acids, polyunsaturated fatty acids, and Hematocrit; rs12767995 (*Arid5b*-expression related) was associated with Rheumatoid arthritis; rs7498665 and others (*Tufm*-methylation related) were linked to a range of traits, including weight, body mass index, etc. The genetic variants linked to secondary traits may introduce horizontal pleiotropy, and further investigations are required to exclude this possibility.

### 3.6. Identification of Drug Properties of Proteins

To validate the potential as drug targets, we also retrieved pertinent drug information for proteins identified through colocalization analysis with a posterior probability greater than 0.7 (PPH4 > 0.7). The results showed that CD274 and FCGR2A proteins have been identified as possible cancer therapeutic targets, indicating the potential and importance of these proteins in anti-cancer drug development (Appendix A).

## 4. Discussion

In this study, we demonstrated that DNA damage repair (DDR) has a causal effect on inflammatory bowel disease (IBD) and its subtypes, and identified important molecules presumed to be involved in IBD development. We identified seven gene expression levels, 33 gene DNA methylation patterns, and two protein expressions that were causally linked to IBD. By integrating data from the drug database, two proteins (CD274 and FCGR2A) were identified that could be used as potential drug targets for IBD or UC. The other genes or methylation sites may serve as potential targets for future small-molecule drug development studies targeting IBD and its subtypes. Furthermore, our study linked genetic loci, DNA methylation, gene expression, and protein expression to IBD, which offers strong evidence for the mechanisms connecting genetic loci, gene expression, methylation, and proteins to IBD.

The expression of seven DDR genes was causally implicated in the development of IBD, including *Arid5b*, *Cox5a*, *Erbb2*, *Ube2l3*, *Gpx1*, *H2bcl2*, and *Mapk3*. We also identified 33 DDR genes whose DNA methylation was causally linked to the development of IBD. Through SMR analysis between mQTL and eQTL, we found that *Erbb2* and *Gpx1* methylation levels may have influenced gene expression. *Erbb2* encodes a receptor tyrosine kinase overexpressing in various tumors [23]. Several studies indicate that *Erbb2* plays a role in regulating the repair of specific DNA damage [24]. It is found in numerous developing mammalian tissues, including the intestinal tract [25]. Our results propose *Erbb2* as a causal gene for UC; the low *Erbb2* expression was correlated with an increased risk of UC. Glutathione peroxidase 1 (GPX1) is a crucial antioxidant enzyme found in the cytoplasm and mitochondria of mammalian cells [26]. The role of *Gpx1* in cellular protection against oxidative stress is pivotal, as it effectively eliminates detrimental peroxides within cells. However, the impact of *Gpx1* can vary significantly depending on distinct physiological and pathological conditions, thereby influencing cell homeostasis maintenance. In multiple cancer types, such as bladder and throat cancers, studies have shown that *Gpx1* expression levels are strongly correlated with prognosis [27,28,29]. *Gpx1* has a preventive effect against the onset and advancement of several chronic illnesses. Studies have shown that *Gpx1* overexpression enhances DNA damage repair capacity [30]. However, under certain circumstances, upregulation of *Gpx1* expression may contribute to cellular dysfunction and disease due to its elimination of crucial reactive oxygen species [26]. Despite its known functions, its causal relationship with IBD and its subtypes remains unclear. Current research indicates that *Gpx1* may exert an influence on the progression of UC. In our SMR analysis conducted on intestinal tissue, we also observed compelling evidence supporting the involvement of *Gpx1* in UC disease progression. However, additional research is warranted to elucidate the exact role of *Gpx1* in the progression of UC, particularly its impact on DNA damage repair mechanisms.

Our study suggests that higher levels of CD274 and FCGR2A protein expression are associated with a reduced risk of IBD and UC. CD274, also known as programmed death ligand 1/PD-L1, is a crucial protein in the immune system, modulating immune responses by binding to PD-1 on T-lymphocytes [31]. In the cytoplasm, PD-L1 indirectly augments DNA damage repair capabilities and confers resistance to DNA damage by safeguarding the mRNA stability of DNA damage-related genes [32]. Previous research has indicated that dysregulation of PD-L1 is associated with various chronic inflammatory diseases [33]. In upper gastrointestinal cancers, the expression of CD274 shows an inverse association with F. nucleatum, a major contributor to colorectal carcinogenesis by inhibiting T-cell-mediated antitumor inflammatory responses [34]. Additionally, some studies suggest that dysregulation of Th1 response in the inflamed colonic mucosa of IBD patients may be linked to alterations in PD-L1 expression in the mucosal stromal compartment [33]. In our study, the low CD274 expression was correlated with an increased risk of IBD, suggesting that the dysfunctional inflammatory responses attributed to CD274 may potentially contribute to the pathogenesis of IBD. Our MR analysis provides further evidence that CD274 is a potential causal protein for IBD. We also used DrugBank to predict that Atezolizumab can target CD274, potentially alleviating the progression of IBD by inhibiting leukocyte infiltration into the intestine and reducing inflammation.

FCGR2A encodes a member of the immunoglobulin Fc receptor gene family, which is ubiquitously expressed on the surface of various immune effector cells [35]. The available data suggest that the FCGR2A plays a pivotal role in the interferon response in systemic lupus erythematosus (SLE), functioning by internalizing immune complexes (ICs) composed of DNA–IgG [36]. FCGR2A-mediated inflammatory signaling can lead to increased oxidative stress, triggering DNA damage and activating DNA repair mechanisms to maintain genomic stability. Studies have shown that the FcgR2a*519G functional variant is associated with IBD and diminishes the susceptibility to UC and CD among Caucasians [37]. Moreover, GWAS in UC has identified a variant in the FCGR2A gene, altering the binding affinity of the encoded antibody receptor, FcγRIIA, for immunoglobulin G (IgG) [38]. In our study, FCGR2A emerges as a promising therapeutic target for IBD, bolstered by compelling causal evidence. This finding not only underscores the pivotal role of DNA damage repair in the pathogenesis of IBD but also lends theoretical support to the development of novel therapeutic strategies. Further investigations will facilitate a deeper understanding of the precise mechanism by which FCGR2A contributes to the pathogenesis of inflammatory bowel disease, as well as explore its potential as a viable therapeutic target for enhancing patient clinical outcomes and quality of life.

IBD encompasses a cluster of chronic disorders characterized by persistent inflammation within the gastrointestinal tract. The pathogenesis of these conditions is intricate, primarily involving an aberrant immune response towards intestinal tissue, resulting in detrimental effects on the intestinal mucosa and perpetuation of an inflammatory cascade. The findings of various studies have demonstrated that chronic inflammatory processes not only exacerbate damage to the intestinal tissue but also induce DNA damage in the intestinal mucosal cells, commonly referred to as intestinal epithelial cells. Under normal circumstances, cellular protective mechanisms are activated upon the occurrence of DNA damage, which includes halting cell division to prevent the accumulation of defective genomes. However, during an inflammatory state, these protective mechanisms may be disrupted, thereby allowing the damaged genome within the cell to persist and accumulate [39]. Genes associated with DNA damage repair play a key role in this process. They are responsible for repairing DNA damage caused by both endogenous (such as metabolites) and exogenous (such as environmental exposures) factors, maintaining the stability of the cell genome to ensure the integrity of genomic variation [40]. In this study, we have identified a causal relationship between the DDR-related molecules CD274 and FCGR2A and IBD. These molecules not only participate in the regulation of immune response and inflammatory processes but also play a pivotal role in the pathological development of IBD. Our recent review article emphasizes the crucial involvement of DDR in modulating the immune microenvironment [41]. Given that abnormal immune and inflammatory responses are the pathological basis of IBD, we speculate that dysfunction of DDR-related molecules leads to alterations in the intestinal immune microenvironment, resulting in the overactivation of immune responses and promoting the onset and progression of IBD.

Recent advancements in inflammatory bowel disease (IBD) treatment, particularly through biological agents and smart biomaterials, have significantly improved patient outcomes. Monoclonal antibodies, such as vedolizumab and ustekinumab, have demonstrated strong efficacy, but their optimal use requires careful management, including dose adjustments, combination therapy with immunomodulators, and therapeutic drug monitoring [42]. Simultaneously, smart biomaterials, like hydrogels and nanoparticles, show promise for targeted drug delivery in IBD, responding to environmental stimuli such as pH and temperature for precise drug release [43]. Despite these advances, challenges remain, including variability in response to monoclonal antibodies, immune tolerance, and side effects, as well as issues with the scalability and stability of smart biomaterials. Ongoing research is needed to refine these therapies and explore novel targets, such as DNA repair and immune modulation, to enhance IBD treatment. Our study possesses several strengths that significantly enhance the robustness and reliability of our findings. Firstly, we employed SMR as our primary analytical method, renowned for its effectiveness in causal inference. Secondly, we conducted colocalization and sensitivity analyses to complement SMR and fortify the validity of our study. This comprehensive approach not only facilitates robust causal reasoning but also ensures the consistency and reliability of our results. Moreover, the utilization of large-scale GWAS data significantly enhances the statistical power of our study. Through analyzing a substantial sample size, we can confidently identify and validate associations between genetic variants and phenotypic traits of interest, ensuring both statistical significance and clinical relevance of our findings. Additionally, our study specifically focuses on individuals with European ancestry to mitigate potential biases arising from diverse genetic backgrounds. This strategic restriction strengthens the internal validity of our results, providing a clearer understanding of how genetic factors influence the observed outcomes in our study population. The combination of SMR methodology, colocalization and sensitivity analyses, large-scale GWAS data utilization, and targeted population selection reinforces the robustness, reliability, and clinical relevance of our study’s findings. These methodological strengths ensure that our conclusions regarding genetic influences on the studied phenotypes are well-supported.

The analysis presented in this study has some limitations as well. Although colocalization analysis effectively accounted for potential bias caused by linkage disequilibrium, we were unable to mitigate the impact of horizontal pleiotropy. Moreover, the GWAS dataset specifically about DNA damage repair is currently unavailable, thereby impeding our ability to employ bi-directional MR analysis using existing software resources for assessing the causal relationship direction. Additionally, we were unable to integrate multi-omics level evidence, including methylation, gene expression, and protein expression, which would have further substantiated the causal association between DDR-related genes and IBD risks. Finally, univariable MR evaluates the overall impact of a single factor on a specific outcome, while multivariable MR extends this analysis by simultaneously assessing multiple potentially correlated factors using the same set of SNPs derived from GWAS summary statistics to determine each factor’s direct influence on the outcome. In this study, DNA damage repair is considered to be a contributing factor that can only be accessed through quantitative trait loci (QTL) datasets rather than GWAS datasets. Therefore, due to data limitations, multivariable MR cannot be employed to analyze the causal relationship between DNA damage repair and IBD and its subtypes. Future research should explore alternative methodologies or utilize available GWAS data to further investigate whether DNA damage repair is causally associated with IBD and its subtypes.

## 5. Conclusions

This study explores the causal relationship between DDR and IBD using multi-omic Mendelian randomization methods. It emphasizes the critical role of DDR genes and their regulatory mechanisms in the pathogenesis of IBD. By delving into these mechanisms, this research enhances our comprehension of IBD pathology and has the potential to identify novel pharmacological targets for treatment, such as CD274 and FCGR2A. Targeting these molecules could complement existing treatments and enhance personalized therapeutic strategies. These findings contribute to advancing medical strategies aimed at managing and possibly preventing IBD more effectively.

## Figures and Tables

**Figure 1 biomedicines-13-00231-f001:**
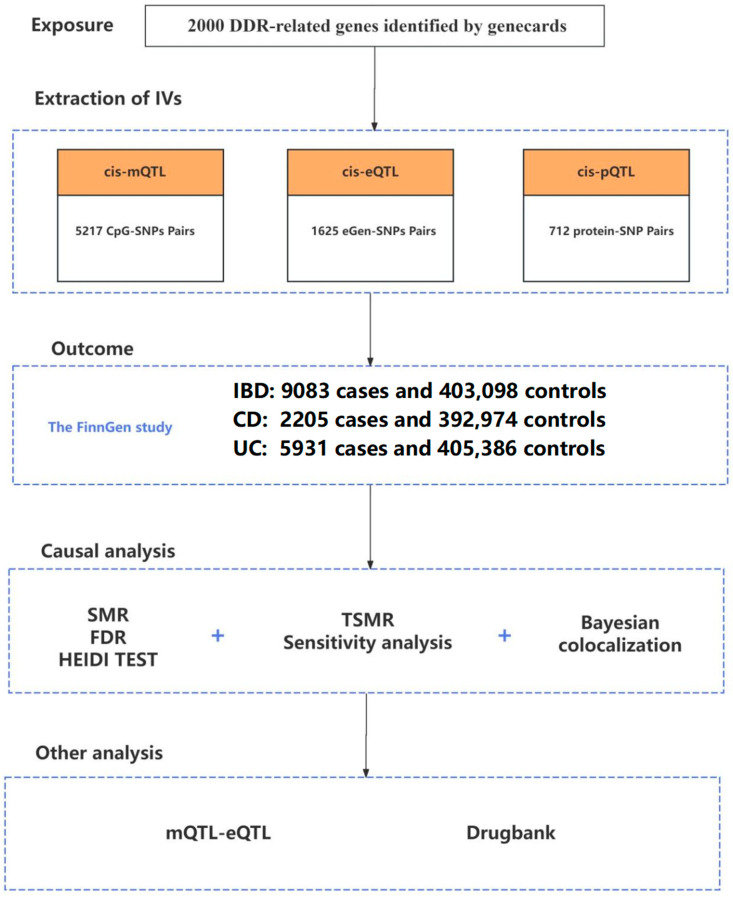
Flowchart of study design and analytical workflow for investigating the causal relationship between DNA damage repair-related genes and inflammatory bowel disease.

**Figure 2 biomedicines-13-00231-f002:**
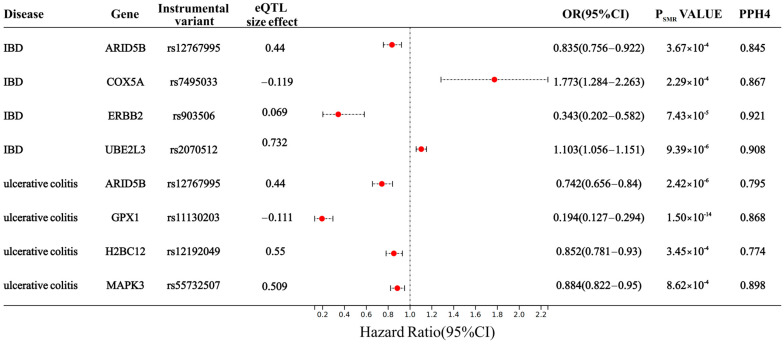
SMR and Colocalization analysis of the DDR-related gene expression with IBD and UC. OR: odds ratio. Colocalization’ denotes the presence of PPH4 between eQTLs and IBD, with a well-established cut-off value of PPH4 > 0.7 indicating strong evidence for colocalization.

**Figure 3 biomedicines-13-00231-f003:**
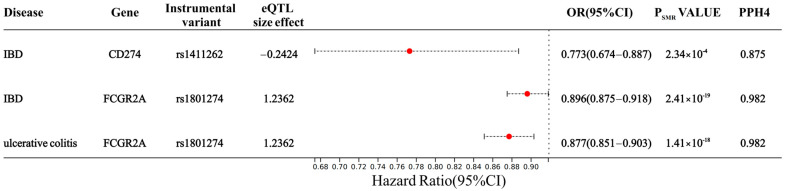
SMR and Colocalization analysis of the DDR-related protein expression with IBD and UC. IBD: inflammatory bowel disease. OR: odds ratio. The term ‘colocalization’ denotes the presence of PPH4 between pQTLs and IBD and UC, with a well-established cut-off value of PPH4 > 0.7 indicating strong evidence for colocalization.

**Table 1 biomedicines-13-00231-t001:** SMR analysis of the DDR-related gene methylation and DDR-related expression.

Outcome	Gene	Probe	eQTL Association			mQTL Association			SMR Association			HEIDI Test	
			β	SE	*p*	β	SE	*p*	β	SE	*p*	*p*	No. of SNPs
IBD ^1^	ERBB2	cg05616858	0.068	0.009	4.60 × 10^−15^	−0.439	0.034	1.16 × 10^−15^	−0.156	0.023	2.27 × 10^−11^	0.263	20
IBD	ERBB2	cg14187895	0.068	0.009	4.60 × 10^−15^	−0.308	0.034	1.71 × 10^−19^	−0.222	0.038	3.24 × 10^−9^	0.141	20
UC ^2^	GPX1	cg24011261	−0.109	0.009	9.86 × 10^−34^	−0.243	0.035	4.76 × 10^−12^	0.450	0.075	1.94 × 10^−9^	0.226	11
UC	GPX1	cg05551922	0.083	0.008	7.44 × 10^−23^	0.193	0.033	3.00 × 10^−9^	0.428	0.084	3.77 × 10^−7^	0.192	7

^1^ IBD, inflammatory bowel disease; ^2^ UC, ulcerative colitis.

## Data Availability

DDR-related genes are available at http://www.genecards.org, accessed on 6 May 2024. GWAS summary statistics for IBD and its two subtypes were down from https://www.finngen.fi, accessed on 10 May 2024. Summary data of mQTL and eQTL were down from https://yanglab.westlake.edu.cn, accessed on 10 May 2024. Summary data of pQTL were down from https://www.decode.com/summarydata, accessed on 10 May 2024.

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
