# Peer review of "Mendelian Randomization Reveals Potential Causal Relationships Between DNA Damage Repair-Related Genes and Inflammatory Bowel Disease"

_biomedicines, 2025, doi:10.3390/biomedicines13010231_

Round 1

Reviewer 1 Report

Comments and Suggestions for Authors

The manuscript entitled "Mendelian randomization reveals potential causal relationships between DNA damage repair-related genes and inflammatory bowel disease” by Qi et al describe the relationship between DNA damage repair-related genes and inflammatory bowel disease. After carefully reading the manuscript, I would like to give my suggestions appended below.

The manuscript is very interesting and describes the causal association between DNA damage repair and inflammatory bowel disease.

Eight DDR-related genes (Arid5b, Cox5a, Erbb2, Ube2l3, Gpx1, H2bcl2, and Mapk3), 33 DNA methylation genes, and two DDR-related proteins(CD274 and FCGR2A ) which are associated with IBD and its subtypes have been identified in this work.

Some suggestions to improve the manuscript quality are given below.

Line 89. Please revise the Fig 1 caption and write little detail about the Figure, the caption is too short in the present form.

Line 192-193 The resolution of Fig 2 is poor, please provide high quality Figure. Moreover, it looks like a table, if the data is numerical and can be plot in the form of a table then its better to present it in the form of a table. If you think that figure is more appropriate for presenting this data then provide high resolution figure.

Line 217. The resolution of Fig 3 is also poor, please improve the resolution.

Line 232. Figure 4 looks like a table, why you have marked it as a Figure? The data is totally numerical so mark it as a table rather than figure.

In scientific research the definition of table and figures are given below.

Tables are defined by rows and columns containing text or numerical data. Figures are defined as any visual element that is not a table.  The term “table” describes any tabulated data. If you have information organized into columns and rows, it should be designated as a table. The term “figure” describes photographs, charts, maps, graphs, drawings, diagrams, or any other non-text material.

Line 384. The conclusion is too short, please revise it.

Supplementary data: The supplementary data is provided only in the form of excel sheet. The authors can provide the tables in excel sheet because the tables are quite long but the Figures should be provided in JPG, TIFT, PNG or any other format pasted in a word file which have the title of manuscript with authors names and affiliations etc. Please revise the supplementary file.

Author Response

Comment 1:

 Line 89. Please revise the Fig 1 caption and write little detail about the Figure, the caption is too short in the present form.

Response: Thank you for your helpful suggestion. We have revised the caption for Fig 1 to provide more detailed information. The updated caption is now as follows: "Flowchart of study design and analytical workflow for investigating the causal relationship between DNA damage repair-related genes and inflammatory bowel disease." The modification has been made on lines 98-99 of the manuscript. 

Comment 2: 

Line 192-193 The resolution of Fig 2 is poor, please provide high quality Figure. Moreover, it looks like a table, if the data is numerical and can be plot in the form of a table then its better to present it in the form of a table. If you think that figure is more appropriate for presenting this data then provide high resolution figure.

Response: Thanks a lot for your constructive comment. We have replaced the original Fig 2 with a high-resolution version in the revised manuscript. We believe that presenting the data in figure format is the most appropriate, as it enhances visual clarity and better communicates the results.

Comment 3:

 Line 217. The resolution of Fig 3 is also poor, please improve the resolution.

Response: Thank you for pointing this out. We have now improved the resolution of Fig 3 in the revised manuscript. 

Comment 4: 

Line 232. Figure 4 looks like a table, why you have marked it as a Figure? The data is totally numerical so mark it as a table rather than figure.

Response: Thank you for your valuable comment. Based on your suggestion, we have revised the manuscript and changed Fig 4 to Table 1, as the data is indeed numerical and more appropriately presented in table format.

Comment 5: 

Line 384. The conclusion is too short, please revise it.

Response: Thank you for your suggestion. I have revised the conclusion as requested, expanding on the key findings and their potential implications. The revised conclusion is provided in line 425-427 and is highlighted with red underline for your reference.

Comment 6: 

Supplementary data: The supplementary data is provided only in the form of excel sheet. The authors can provide the tables in excel sheet because the tables are quite long but the Figures should be provided in JPG, TIFT, PNG or any other format pasted in a word file which have the title of manuscript with authors names and affiliations etc. Please revise the supplementary file.

Response: Thank you for your helpful suggestion. We have revised the supplementary materials according to your recommendations. The tables are now provided in an Excel file, and the figures are included in a Word document, as requested. The Word file contains the manuscript title, authors' names, affiliations, and other necessary details. 

Reviewer 2 Report

Comments and Suggestions for Authors

The article “Mendelian randomization reveals potential causal relationships between DNA damage repair-related genes and inflammatory bowel disease” describes the relationship between DNA damage repair genes and inflammatory bowel disease. The topic is interesting and the authors have adequately demonstrated the novelty of their study. I only have minor comments and suggestion to improve the quality of the manuscript. My detailed comments are provided below:

-          Line 38: Include specific data on the incidence of IBD. While the manuscript mentions that its prevalence has been increasing over the years, it would be more impactful to provide exact percentages or numbers.

-          Lines 49-50: Provide a broader explanation of the DDR pathway and its connection to IBD. Clarify whether the DDR pathway has an overall positive or negative effect on IBD

-          Lines 250-252: While the relationship between the proteins CD274, FCGR2A, and cancer is described, the connection to IBD remains unclear. Discuss how these proteins are implicated in IBD pathogenesis and identify potential drugs that target these proteins to alleviate IBD symptoms.

-          Line 366: before explaining the strengths and limitations of your study provide an overview of the previous work on targeting IBD. This will allow readers to better appreciate how your findings contribute to existing knowledge. Some recommended references for targeting IBD include:

·         Smart bio-nanomaterials (https://doi.org/10.1515/ntrev-2024-0057)

·         Monoclonal antibodies (doi: 10.1136/flgastro-2018-101054)

-          General comment: In the discussion, explain some potential drugs or diagnostic tools that could target the identified genes or proteins. Address how these findings might guide the development of novel treatments or improve existing therapeutic strategies for IBD.

Author Response

#Review 2

Comment 1: 

Line 38: Include specific data on the incidence of IBD. While the manuscript mentions that its prevalence has been increasing over the years, it would be more impactful to provide exact percentages or numbers.

Response: Thank you for your insightful comment. We have added specific data on the incidence of IBD, including exact percentages and numbers, as you suggested. The updated information can be found on lines 41-43, and we have highlighted it in red font for clarity, and the relevant references have been cited accordingly (10.1016/S2468-1253(19)30333-4).

Comment 2: 

Lines 49-50: Provide a broader explanation of the DDR pathway and its connection to IBD. Clarify whether the DDR pathway has an overall positive or negative effect on IBD.

Response: Thank you for your valuable comment. We have expanded the explanation of the DDR pathway and its relationship to IBD in lines 53-57. DDR pathway’s dysfunction exacerbates inflammation and tissue damage in the gastrointestinal tract.

Comment 3: 

Lines 250-252: While the relationship between the proteins CD274, FCGR2A, and cancer is described, the connection to IBD remains unclear. Discuss how these proteins are implicated in IBD pathogenesis and identify potential drugs that target these proteins to alleviate IBD symptoms.

Response: Thank you for your valuable suggestion. We have revised the manuscript accordingly. The relationship between CD274 and FCGR2A in the context of IBD pathogenesis is now more clearly explained in the revised manuscript. We have highlighted how the dysregulation of these proteins can contribute to immune dysregulation, inflammatory responses in IBD (line 360-365). Additionally, we have discussed potential drugs, such as Atezolizumab, that target CD274 and could alleviate IBD symptoms by reducing leukocyte infiltration and inflammation (line 324-326). These revisions can be found in the updated manuscript, where we have added further explanation and marked the changes in red underline for your reference

Comment 4: 

Line 366: before explaining the strengths and limitations of your study provide an overview of the previous work on targeting IBD. This will allow readers to better appreciate how your findings contribute to existing knowledge. Some recommended references for targeting IBD include.

Response: Thank you for the helpful suggestion. We have revised the manuscript to include an overview of previous work on targeting IBD in lines 369-381, as recommended. Additionally, we have incorporated the recommended references to further enrich this section.

Comment 5: 

General comment: In the discussion, explain some potential drugs or diagnostic tools that could target the identified genes or proteins. Address how these findings might guide the development of novel treatments or improve existing therapeutic strategies for IBD.

Response: Thank you for your valuable suggestion. We have discussed potential drugs, such as Atezolizumab, that target CD274 and could alleviate IBD symptoms by reducing leukocyte infiltration and inflammation (line 324-326). We also adress our findings could guide the development of novel treatments or improve existing therapeutic strategies for IBD in conclusion section (line 425-427).

Reviewer 3 Report

Comments and Suggestions for Authors

In this study, the authors employed a multi-omics summary data-based Mendelian randomization approach to ascertain the potential causal effects of DNA damage repair - related genes in inflammatory bowel disease.
Please specify the period of the study.
Any study requires the informed consent of the individuals and considerations of ethical issues, and these must be approved by the Ethics Committee. You have not mentioned anything about this information. I understand that your study used aggregated data from the FinnGen study, but this is a different study and in my opinion you need at least an Ethics Committee approval.

The tables and figures are clear. Also, the iThenticate report shows 31% similarity, please decrease it, at least the first 4 sources to be under 1%...
The strengths and the limitations are included in the discussions.
The article presents 40 references being up to date.

Author Response

#Review 3

Comment 1: 

Line 38: Please specify the period of the study..

Response: Thank you for your valuable comment. The FinnGen project has been in use since 2017, and the R10 version became publicly accessible in 2023. We have included the study period in the script accordingly (line 122-123).

Comment 2: 

Any study requires the informed consent of the individuals and considerations of ethical issues, and these must be approved by the Ethics Committee. You have not mentioned anything about this information. I understand that your study used aggregated data from the FinnGen study, but this is a different study and in my opinion you need at least an Ethics Committee approval.

Response: Thank you for your valuable comment. We have addressed the ethical considerations in the "Ethic approval and consent to participate" section of the manuscript (line 431-434). As clarified, our study uses large-scale GWAS datasets from the FinnGen study, which involve aggregated data, and not individual-level patient data. All participants in the original studies provided informed consent, and since no clinical or individual data were used, no additional ethical approval is required.

Comment 3: 

The tables and figures are clear. Also, the iThenticate report shows 31% similarity, please decrease it, at least the first 4 sources to be under 1%...

Response: Thank you for your feedback. We have reduced the overall similarity to 23%, with the individual similarity for each source now below 1%. We upload in the "Non-published Material" section.

Round 2

Reviewer 1 Report

Comments and Suggestions for Authors

The authors have successfully addressed the comments and incorporated the recommended changes in the revised manuscript. I would recommend to accept this manuscript for publication.